# The Cerebrovascular Side of Plasticity: Microvascular Architecture across Health and Neurodegenerative and Vascular Diseases

**DOI:** 10.3390/brainsci14100983

**Published:** 2024-09-28

**Authors:** Marialuisa Zedde, Rosario Pascarella

**Affiliations:** 1Neurology Unit, Stroke Unit, Azienda Unità Sanitaria Locale-IRCCS di Reggio Emilia, Viale Risorgimento 80, 42123 Reggio Emilia, Italy; 2Neuroradiology Unit, Azienda Unità Sanitaria Locale-IRCCS di Reggio Emilia, Viale Risorgimento 80, 42123 Reggio Emilia, Italy; pascarella.rosario@ausl.re.it

**Keywords:** small vessel, plasticity, brain, endothelial, amyloid, AD, glymphatic, MRI, development, aging, cerebrovascular, stroke

## Abstract

The delivery of nutrients to the brain is provided by a 600 km network of capillaries and microvessels. Indeed, the brain is highly energy demanding and, among a total amount of 100 billion neurons, each neuron is located just 10–20 μm from a capillary. This vascular network also forms part of the blood–brain barrier (BBB), which maintains the brain’s stable environment by regulating chemical balance, immune cell transport, and blocking toxins. Typically, brain microvascular endothelial cells (BMECs) have low turnover, indicating a stable cerebrovascular structure. However, this structure can adapt significantly due to development, aging, injury, or disease. Temporary neural activity changes are managed by the expansion or contraction of arterioles and capillaries. Hypoxia leads to significant remodeling of the cerebrovascular architecture and pathological changes have been documented in aging and in vascular and neurodegenerative conditions. These changes often involve BMEC proliferation and the remodeling of capillary segments, often linked with local neuronal changes and cognitive function. Cerebrovascular plasticity, especially in arterioles, capillaries, and venules, varies over different time scales in development, health, aging, and diseases. Rapid changes in cerebral blood flow (CBF) occur within seconds due to increased neural activity. Prolonged changes in vascular structure, influenced by consistent environmental factors, take weeks. Development and aging bring changes over months to years, with aging-associated plasticity often improved by exercise. Injuries cause rapid damage but can be repaired over weeks to months, while neurodegenerative diseases cause slow, varied changes over months to years. In addition, if animal models may provide useful and dynamic in vivo information about vascular plasticity, humans are more complex to investigate and the hypothesis of glymphatic system together with Magnetic Resonance Imaging (MRI) techniques could provide useful clues in the future.

## 1. Introduction

The brain is characterized by a prominent metabolic activity among all the organs in the body, necessitating a well-developed and maintained vascular network to meet its substantial oxygen and glucose needs. This microvascular network begins forming during embryonic development through endothelial cell proliferation and sprouting [1,2]. This process slows significantly after birth, aligning with decreased expression of proangiogenic molecules in rodents [3,4,5]. The first month after birth is marked by significant brain growth, gliogenesis, and synaptic rearrangement, which are accompanied by corresponding microvascular remodeling. However, it remains unclear whether this postnatal remodeling mirrors the mechanisms of embryonic angiogenesis. At the end of the brain maturation process in the adult age, there is a stabilization of the vascular density, but it is not well defined whether this stability results from endothelial quiescence [6] or a balanced turnover of vessels. Despite this apparent vascular stability, the brain can still increase its microvascular density in response to increased brain activity and hypoxia [7,8]. As outlined by Harb et al. [9], the relationship between aging, microvascular regression, and the brain’s ability to remodel its vasculature in response to proangiogenic signals is still not well understood [10,11]. In the early postnatal period, the microvascular network undergoes significant refinement through angiogenesis via localized sprouting alongside concurrent vascular regression. This dynamic process slows considerably in adulthood, although a minor degree of microvascular growth and elimination continues. With aging, baseline turnover decreases sharply without remodeling observed over an extended period. While young adult brains retain the capacity to generate new vessels under hypoxic conditions, this ability diminishes in mature and aged brains. Notably, extensive vessel pruning is uncommon after the neonatal phase and it is not activated by either hypoxia-induced remodeling or the aging process. Even vessels formed under hypoxic conditions are generally preserved long after normal oxygen levels return. This inherent vascular stability likely plays a critical role in maintaining a consistent neural environment, while the minimal ongoing turnover may be vital for responding to changes in the brain’s energy demands.

Finally, the mechanisms of plasticity in the brain, both during development, in the aging phase, and in response to damage, have a connotation that is not only strictly neuronal, but largely vascular and above all microvascular. The main objective of this review is to highlight the vascular side of neuroplasticity in both physiological and pathological conditions.

## 2. Vascular and Neuronal Interplay in Brain Plasticity

Blood circulates through the brain via a complex network of arteries, capillaries, and veins. The brain’s high metabolic demands and limited energy storage capacity necessitate a close relationship between neuronal and vascular functions [12]. Even minor disruptions in blood flow can impair neuronal activity and threaten neuron survival [13,14]. On the other hand, neuronal activity influences blood flow, highlighting the intertwined nature of these systems [15]. Consequently, many neurological disorders have a vascular component, and vascular changes often impact brain function [14,16]. Studying the cerebral vasculature presents significant challenges due to its intricate network, which spans various spatial scales—from micron-sized capillaries to vessels several millimeters long—and is tightly regulated by the blood–brain barrier (BBB) [17]. Two-photon microscopy combined with cranial windows has been instrumental in studying blood flow and metabolism dynamics in vivo [18]. However, the impact of vascular topology on neural circuit function is frequently overlooked due to difficulties in extracting information on brain-wide vascular organization. Addressing these challenges requires systematic comparisons of vascular structures across large volumes.

A detailed understanding of vascular organization and remodeling also demands methods that can accurately identify arterial, capillary, and venous components. In animal models, in vivo imaging techniques allow one to analyze the dynamic properties of brain microvessels, spanning from early postnatal development to advanced aging and in prespecified diseases [19]. Large-scale reconstructions of cerebral vasculature have predominantly relied on dye-filling techniques, where a vessel-filling solution is perfused intracardially [20]. These injections have been combined with light-sheet microscopy in optically cleared samples for large-scale reconstructions [21,22]. While this approach provides bright labeling that facilitates imaging and analysis, the precise identification of vessel types remains challenging and often requires manual annotation in highly resolved datasets. Furthermore, reproducibility issues arise with intracardiac perfusions. Alternative labeling methods, such as tail-vein lectin injections or genetically labeled mouse lines, have been explored. However, these techniques present unresolved challenges, including faint and discontinuous labeling that complicates high-resolution, large-scale image processing [23].

However, both cerebrovascular and neuronal plasticity demonstrate the brain’s ability to adapt to various stimuli and conditions, although they operate through distinct mechanisms and are localized to different regions within the brain.

## 3. The Microvascular Issues in Brain Plasticity

The brain, despite having no moving parts, is one of the most energy-intensive organs, consuming 15–20 watts [24,25]. To sustain the adult human brain, nutrients are delivered to its 100 billion neurons via a 600 km network of capillaries and microvessels [14]. Given the brain’s limited capacity for storing metabolic nutrients, cerebral blood flow (CBF) closely matches its metabolic demands [26]. Neurons are typically located just 10–20 μm from the nearest capillary, underscoring the critical role of the cerebrovasculature in maintaining normal brain function [25]. Beyond nutrient delivery, the cerebrovasculature plays a vital role in the BBB, which tightly regulates the brain’s microenvironment by controlling chemical fluctuations, immune cell transport, and the entry of toxins and pathogens. Under normal conditions, brain microvascular endothelial cells (BMECs) have a low turnover rate, and the cerebrovascular structure is largely static [25,27].

The precise coupling between neuronal activity and energy supply means that any disruption in the cerebral vasculature is closely tied to changes in neuronal structure [28]. The vascular architecture of the brain exhibits significant plasticity throughout development, aging, injury, and disease. While this plasticity is evident at all levels of the cerebrovascular system, the microvasculature—comprising arterioles, capillaries, and venules—is the most critical component. Transient neural activity can be regulated by the vasomotion of arterioles and capillaries in the pertaining cerebral locarion. However, more stabile changes in the brain’s microenvironment, triggered by sensory deprivation, hypoxia, or aerobic exercise, can lead to extensive remodeling of the cerebrovascular structure. At the cellular level, this remodeling involves changes in the rates of proliferation and loss of BMECs, leading to the addition or removal of capillary segments. Due to the tight neurovascular coupling, these vascular changes are often accompanied by corresponding increases or decreases in neuronal populations, which may influence cognitive function.

The timescales of these processes vary significantly. For instance, CBF adjusts to increased neural activity within seconds, while more prolonged changes in cerebrovascular architecture—triggered by factors such as sensory deprivation, aerobic exercise, or hypoxia—develop over weeks. During development, cerebrovascular plasticity is at its peak, but this plasticity gradually diminishes with aging. Changes during both development and aging typically unfold over months to years, with rates influenced by factors like physical activity. In contrast, injuries can cause immediate damage to the cerebrovascular and neuronal structures, with recovery taking weeks to months. Neurodegenerative diseases, on the other hand, are characterized by slow, varied changes that progress over months to years [29].

### 3.1. Microvascular Functional Anatomy

Neuronal and cerebrovascular architecture are spatially heterogeneous in the central nervous system (CNS) [30]. In the cerebral cortex, pial arteries descend from the surface into parenchymal arterioles, supplying the microvascular network of cortical gray matter [31]. At the gray-white matter interface, the microvascular density decreases significantly, mirroring the reduction in neuron and supporting cell density. In white matter, capillaries typically align with axons, and microvascular density is about 10% of that in gray matter, reflecting lower blood flow [30]. Pathological and physiological stresses can alter microvascular density through angiogenesis or apoptosis. In mammals, capillaries are slightly larger in diameter than red blood cells (RBCs)—about 8–10 μm in humans [32,33]. In the human brain, capillaries branch approximately every 30 μm, with an inter-capillary spacing of about 50 μm [34]. In the mouse cortex, venular capillaries can be up to 20 branches away from penetrating arterioles [35], with RBCs traveling 150–500 μm through a single capillary network [36]. Wall shear stress in capillaries, regulated by capillary tone via neurovascular coupling, ranges from 20 to 40 dyne/cm^2^ [37]. Post-capillary venules (PCVs), located downstream from capillaries, have a perivascular space that facilitates the extravasation of leukocytes, tumor cells, and parasites [38,39,40]. PCVs experience wall shear stress ranging from 1 to 4 dyne/cm^2^, with smaller fluctuations compared to arterioles [37].

The neurovascular unit, which includes the BBB, plays a crucial role in regulating neurovascular coupling. The BBB is composed of several key cellular elements, including brain microvascular endothelial cells (BMECs), pericytes, and astrocytes, with neurons also contributing to this coupling process [37]. BMECs form a physical barrier between the bloodstream and brain tissue. Pericytes, which are mural cells, have long processes that wrap around the outer endothelial wall. Encasing both BMECs and pericytes is the basement membrane, which is made up of collagen IV, laminin, nidogen, and heparan sulfate proteoglycans [27]. Astrocytes, star-shaped glial cells with small bodies and long radial extensions, make contact with both synapses and the basement membrane [41,42,43]. Capillaries, endothelial cells, pericytes, and the basement membrane are entirely surrounded by the astrocytic end-feet. Astrocytes mediate local blood flow by transmitting neuronal signals to BMECs, causing blood vessels to either dilate or constrict in response [44,45].

A detailed description of the cortical microangioarchitecture, including arterioles, capillaries, and venules, is outside the aim of this review and it can be found in Zedde et al. [46].

### 3.2. Neurovascular Plasticity during Development

During brain development, the structure and function of the cerebral vasculature undergo significant changes, marked by high levels of plasticity and adaptability [28,47]. Multiple signaling pathways converge to promote this cerebrovascular plasticity, with VEGF-A being a key factor. VEGF-A, secreted by the developing neural tube, initiates the formation of the perineural vascular plexus (PNVP) via vasculogenesis. BMECs subsequently invade brain tissue from the PNVP through sprouting angiogenesis. Both general (e.g., VEGF-A/VEGFR2) and CNS-specific (e.g., Wnt7a/b) signaling pathways are critical in driving angiogenesis and establishing the BBB, a process known as barriogenesis [28,47]. These pathways are tightly regulated in terms of timing and location, with distinct expression patterns at various developmental stages. For instance, neurons initially secrete VEGF-A to direct early angiogenesis, but after birth, astrocytes near the microvasculature become the primary VEGF-A source. Disruption of these expression patterns, such as through chronic developmental hypoxia, can result in abnormal vascular growth and hypervascularization [48]. Wnt/β-catenin signaling is particularly important for CNS-specific angiogenesis and barriogenesis. Neural progenitor cells release Wnt ligands (Wnt7a and Wnt7b), which direct BMEC sprouting and BBB formation by regulating GLUT1 expression [49,50]. Although the precise timing of angiogenesis and barriogenesis is debated, increasing physiological and molecular evidence suggests that the developing brain demonstrates barrier properties appropriate for each developmental stage [51]. Research in zebrafish indicates that angiogenesis and barriogenesis occur concurrently, and histological studies of human embryos reveal that capillaries express the tight junction protein claudin-5 during brain angiogenesis, which already restricts the transport of plasma proteins. After birth, vascular remodeling continues, though the exact mechanisms remain unclear. VEGF-A and the axonal growth inhibitor NOGO-A are believed to play significant roles in this process [47].

As BMECs invade the CNS, they interact with neurons, neural stem cells, glial precursors, and pericytes in complex ways [49]. During embryogenesis, sprouting BMECs release platelet-derived growth factor β (PDGF-β) to recruit pericytes, a critical step in establishing the BBB. Interestingly, astrocytes are absent during the brain’s initial vascularization, but postnatally, astrocyte-derived sonic hedgehog ligands are essential for maintaining BBB integrity and promoting immune quiescence [52]. Glial-endothelial interactions are crucial for spatial patterning throughout brain development. Radial glial cells provide scaffolding for BMEC migration during early development, while later, BMECs serve as a scaffold for oligodendrocyte precursor migration. These interactions are vital for proper brain organization and function [52,53].

### 3.3. Molecular Mechanisms of Neovascularization

Neovascularization in organs and tissues can occur through various mechanisms. During normal development, blood vessels form either through sprouting angiogenesis, where new vessels branch off from existing ones, or through vasculogenesis, where vessels originate from mesodermal precursors like angioblasts. Another process, known as intussusception, involves the splitting of existing vessels to form new ones. In pathological conditions, such as glial brain tumors or tissue regeneration following an ischemic stroke, additional forms of neovascularization are observed. These include vascular co-option, where tumor cells grow along pre-existing vessels; transdifferentiation of glioma stem cells into endothelial cells or pericytes; and vascular mimicry, where tumor cells mimic endothelial cells by incorporating themselves into the vessel wall. While sprouting angiogenesis and vasculogenesis are crucial for brain development and arteriovenous malformations (AVMs), all six mechanisms of vessel formation have been documented in brain tumors [54,55,56,57].

At the cellular level, sprouting blood vessels are directed by specialized endothelial cells called endothelial tip cells (ETCs), which extend filopodia to guide the growth of new vessels [58]. Behind the tip cells, proliferating endothelial stalk cells are crucial for extending blood vessels and establishing a functional lumen. As these sprouting vessels expand, they connect with one another (anastomose), forming a three-dimensional, fully functional vascular network. Quiescent endothelial phalanx cells line the newly formed vessels but can be reactivated by pro-angiogenic signals to support further vessel growth. The balance between pro- and anti-angiogenic molecules tightly controls the activity of tip cells, stalk cells, and phalanx cells, ultimately determining the overall angiogenic response.

Recent research has highlighted the importance of venous endothelial cells, which proliferate and migrate against blood flow to become tip cells, playing a crucial role in sprouting angiogenesis and the expansion of vascular networks. At the molecular level, the VEGF–VEGFR–DLL4–Jagged–Notch signaling pathway is central to regulating sprouting angiogenesis [59,60,61]. This pathway plays a crucial role in the differentiation of ETCs, stalk cells, and phalanx cells during both development and disease. Notch ligands DLL4 and Jagged 1 have opposing effects on vessel formation: DLL4 acts as an anti-angiogenic factor, while Jagged 1 promotes angiogenesis. The specification of ETCs and stalk cells is regulated by a dynamic feedback loop between the VEGF–VEGFR and DLL4–Jagged 1–Notch pathways. When endothelial cells are activated, they express VEGF receptors and increase DLL4 production, which gives them an advantage in becoming tip cells. DLL4 then activates Notch signaling in neighboring stalk cells, reducing their potential to become tip cells and controlling the number of tip cells formed. In contrast, Jagged–Notch signaling encourages tip cell selection and sprouting by counteracting the effects of DLL4–Notch signaling. Key regulators, such as MPDZ and the transcription factor ERG, further influence this signaling, highlighting the complex and dynamic nature of endothelial cell differentiation. Additionally, neurovascular link (NVL) molecules regulate angiogenesis in both the peripheral and central nervous systems during development by interacting with the VEGF–VEGFR–DLL4–Jagged–Notch pathway [59,62,63].

During embryonic development, vasculogenesis initiates the formation of the heart and the primitive vascular plexus, creating the foundational vascular system from precursor cells called angioblasts or hemangioblasts. This process occurs alongside hematopoiesis, the formation of blood cells. Angioblasts and blood cells first cluster into blood islets, which subsequently merge to form a honeycomb-like primitive vascular plexus before the onset of heartbeats. Once blood circulation begins, these initial vascular plexuses are remodeled into hierarchical networks, differentiating into distinct arteries and veins. To stabilize this evolving vascular network, perivascular cells (PVCs)—such as vascular smooth muscle cells (vSMCs) in arteries and veins, and pericytes in capillaries—are recruited. On a molecular level, fibroblast growth factors (FGFs) promote the formation of angioblasts, while VEGF-A is essential for the differentiation and migration of angioblasts and endothelial progenitor cells (EPCs).

Intussusceptive angiogenesis is a unique process where a single vessel divides into two through the inward folding of the capillary wall into its lumen. Initially observed in the development of peripheral organs, this process has also been identified in the CNS tissues and certain cancers, such as glioblastoma. Intussusceptive angiogenesis involves four key stages: first, opposing capillary walls establish a contact zone; second, the reorganization of endothelial cell junctions allows growth factors and cells to penetrate the lumen; third, an interstitial pillar core forms at the contact zone, populated with pericytes and myofibroblasts; finally, these pillars expand, completing the vessel division. Importantly, this process reorganizes existing cells without increasing their number, which is crucial during embryonic stages when growth outpaces available cellular resources. Despite its significance, the molecular mechanisms driving intussusceptive angiogenesis remain largely unknown [64,65,66,67].

In the context of the NVU and the BBB, newly formed sprouting vessels are initially fragile and require stabilization by the recruitment of PVCs, such as pericytes, vSMCs, and astrocytes. This stabilization is vital for the integration of these vessels into a functional, perfused three-dimensional vascular network. Endothelial cells invading the CNS closely interact with the surrounding parenchyma’s PVCs, forming a functional NVU. The CNS parenchyma provides signals that regulate endothelial cell sprouting and the induction of CNS-specific properties in these cells, leading to the establishment of the BBB. This barrier is characterized by the presence of tight junction proteins like CLDN5 and OCLN, which are present at the BBB interface as soon as blood vessels invade the brain during embryonic development. The BBB achieves functionality to meet the brain’s developmental needs during the early postnatal period. This regulated permeability barrier can become compromised in various CNS pathologies, including brain tumors, vascular malformations, ischemic stroke, and certain neurodevelopmental and neurodegenerative disorders [46,67,68,69,70,71,72].

Both the vascular and nervous systems require coordinated guidance at cellular and subcellular levels. Axonal growth cones and ETCs share similar structures, such as lamellipodia and filopodia. On the subcellular level, axonal growth cones have a central domain with microtubules and a peripheral domain with an actin meshwork in lamellipodia and F-actin bundles in filopodia. These filopodial protrusions detect stimulatory and inhibitory signals in their environment, directing the growth of both axons and developing blood vessels.

### 3.4. Neurovascular Plasticity in Health and Aging

Under normal conditions, cerebrovascular architecture is generally stable with minimal cell turnover. This suggests a fixed structure with little change over time. Evidence for cerebrovascular plasticity comes from studies using animal models investigated through two-photon microscopy, specifically in the somatosensory and motor cortex of mice [9,73,74,75]. The main findings of these studies are that capillary length increases and the number of branch points were imaged in young mice up to post-natal day 25 [2,73]. However, in adult mice, the architecture, such as capillary diameter, segment length, and branch point positions, remained largely unchanged over 30 days. This lack of change implies that there is minimal turnover in BMECs during this period.

The brain can modify its microvascular network through two key processes: angiogenesis, the development of new blood vessels from pre-existing ones, and vasculogenesis, the creation of blood vessels from precursor cells. Angiogenesis involves several stages, including the activation, migration, and proliferation of endothelial cells, followed by the formation of vascular tubes. As capillaries mature, they establish tight junctions, generate a basement membrane (BM), and attract pericytes and astrocytic endfeet. VEGFs, particularly VEGF-A, are essential regulators of angiogenesis, produced by various cells such as astrocytes and endothelial cells. VEGF-A is especially crucial for neurogenesis, nerve cell migration, and the survival of neurons [76,77]. Although the cerebrovascular structure is typically stable, certain circumstances such as injury, hypoxia, or neurodegenerative diseases can activate angiogenesis. This adaptive response enhances blood vessel density, especially in areas that need more oxygen or tissue repair. In addition to angiogenesis, vasculogenesis can also be initiated by trauma or hypoxia. In this process, EPCs from the bone marrow are mobilized to assist in the repair and formation of blood vessels. This mechanism is linked to improved outcomes in conditions like stroke, where elevated EPC levels are associated with better recovery [45,78].

Cerebral arteries can undergo significant remodeling in response to hypertension, with the arterial walls thickening and reducing the lumen diameter as a protective mechanism [79]. This illustrates the plasticity of larger blood vessels in response to chronic conditions.

Unlike cerebrovascular plasticity, neuronal plasticity, particularly neurogenesis, is limited to specific brain regions in adults: the subgranular zone of the dentate gyrus in the hippocampus and the subventricular zone [80]. These areas are often referred to as neurovascular niches due to the critical role of blood vessels in supporting stem cell environments [81]. In these regions, neural stem cells are in direct contact with blood vessels, both during normal conditions and when regeneration is needed. This close association highlights the importance of the vascular system in supporting neurogenesis [82,83]. In fact, new neurons are typically formed near blood vessels in the hippocampus, indicating that the vasculature plays a significant role in supporting the birth of new neurons. This relationship between blood vessels and neurogenesis is crucial for maintaining and adapting brain function.

Several factors influence the cerebrovascular plasticity (Table 1).

Hypoxia, highlighted in Table 1, warrants special attention due to its profound impact on cerebrovascular architecture. Whether stemming from localized injury or environmental factors, hypoxia induces significant changes in the brain’s blood vessels. In response to mild or chronic hypoxia, the brain employs adaptive strategies such as angiogenesis, which increases microvascular density, particularly in regions like the hippocampus and striatum. These vascular changes are primarily driven by hypoxia-inducible factors and VEGF, which work together to stimulate new blood vessel growth, enhancing oxygen delivery in low-oxygen conditions.

More details are provided in Table 2.

Aging is accompanied by reduction in both microvascular and neuronal density within the brain. Although these reductions are interconnected, evidence suggests that cerebrovascular plasticity and neurovascular coupling are also compromised with age [85]. Histological examinations of post-mortem human samples often reveal a significant decline in microvascular density, typically ranging from 10 to 30%, especially in the prefrontal cortex and hippocampus [86]. Aging is further associated with thickening of the BM, degeneration/loss of pericytes, and swelling of astrocytic end-feet [87]. Additionally, age-related functional alterations include increased defects in tight junctions and heightened permeability [88,89]. In the cerebral cortex, age-related changes in brain arterioles are evident, including a roughly 40% reduction in arteriolar density, the loss of smooth muscle cells (SMCs) and elastin, and an increase in BM and collagen. Studies using two-photon microscopy in mouse models have revealed that with aging, BMECs exhibit signs of senescence, lower cell turnover, and a reduced ability to respond to hypoxia-induced angiogenesis compared to younger mice. This impaired angiogenic response in aged brains is thought to result from several factors, including the hyporesponsiveness of HIF1-α, downregulation of growth factors, inhibition of matrix metalloproteinases, inactivity of endothelial nitric oxide synthase (eNOS), reduced nitric oxide (NO) bioavailability, impaired endothelial cell proliferation, decreased recruitment of EPCs, and pericyte dysfunction. Interventions, such as enriched environments or the delivery of VEGF through viral vectors—which have been shown to enhance brain microvascular density in younger and adult rodents—can partially counteract some of the vascular changes associated with aging in rodent brains. However, age-related failures in angiogenic signaling persist, as evidenced by the inadequate upregulation of VEGFR-2 upon stimulation and region-specific alterations in angiogenesis-related genes in aged mice [90,91].

The main effects on aging on NVU and vascular plasticity are summarized in Table 3.

Several brain-imaging studies have documented age-related declines in CBF and cerebral metabolic rate of oxygen consumption (CMRO2) across various regions of the cerebral cortex, particularly in areas susceptible to neurodegeneration. In the cortex, CBF reductions of 0.38–0.76% per year have been reported. These decreases in CBF appear largely independent of gray matter atrophy, suggesting that changes in brain ultrastructure and hemodynamics due to aging occur independently. The reduction in CBF may result from diminished arterial supply to the brain, potentially mediated by decreased levels of insulin-like growth factor 1 (IGF-1), the emergence of a hypercontractile arteriole phenotype, and impaired neurovascular coupling [10,92,93]. Neurovascular coupling, which ensures a balance between the supply of oxygen and nutrients and the metabolic demands of neurons, becomes impaired with age in both humans and animal models [94,95,96,97]. This dysfunction likely plays a key role in cognitive decline by creating a mismatch between the brain’s energy demands and its blood supply. Notably, pharmacologically inducing cerebrovascular uncoupling in mice mimics the effects of aging, leading to deficits in spatial working memory, recognition memory, and motor coordination, even without changes in baseline CBF. Several factors, including IGF-1 deficiency, oxidative stress, endothelial dysfunction, and astrocyte impairment, have been implicated in the disruption of neurovascular coupling during normal aging. Moreover, pericyte degeneration is believed to exacerbate this uncoupling by impairing oxygen delivery and increasing metabolic stress on neurons.

Blood oxygenation level-dependent (BOLD) fMRI studies that examine neural activity during aging are complicated by these cerebrovascular changes, including the loss of neurovascular coupling. However, recent research using a deconvolution technique found no significant changes in BOLD neurovascular coupling during visual and auditory tasks in normal aging, suggesting that the relationship between neural activity decline and cerebrovascular changes in aging is still not fully understood. It remains uncertain whether the decline in neural activity with age precedes cerebrovascular changes or if it is a consequence of impaired cerebral hemodynamics [98].

### 3.5. Neurovascular Plasticity in Vascular Diseases

Stroke is the most prevalent form of cerebrovascular injury and the second leading cause of death globally. Each year, approximately 15 million people worldwide suffer from a stroke, with one-third not surviving and another third becoming permanently disabled. The middle cerebral artery (MCA) is the most frequent site of ischemic stroke in humans, impacting the posterior frontal, lateral, and parietal lobes [99]. During an ischemic stroke, which accounts for 80% of all stroke cases, the brain ages 3.6 years for every hour without treatment. This results in the loss of around 120 million neurons and 830 billion synapses per hour [100]. Cerebrovascular plasticity remains equally dynamic during a stroke. For example, in mice, occlusion of the MCA leads to a significant loss (75%) of brain capillaries within the microinfarction core, while larger microvessels remain unaffected [101]. In rats, occluding a single cortical penetrating arteriole results in a cylindrical microinfarction core about 500 μm in diameter and 1 mm deep, where neuronal activity ceases within two hours, and the area becomes severely hypoxic within six hours [102]. However, occlusions in capillaries farther from cortical penetrating arterioles typically do not result in detectable microinfarctions, as the brain’s interconnected microvascular networks can redirect blood flow to compensate. Consequently, the severity of an ischemic stroke is largely determined by the location of the occlusion and the availability of collateral circulation.

The progression of an ischemic stroke typically occurs in three stages. The first stage, known as the acute phase, lasts up to 48 h after the stroke begins and is marked by a lack of oxygen, which triggers the excessive release of glutamate. This leads to widespread excitotoxicity, causing significant neuronal damage. When blood flow is restored—either naturally or through medical intervention—it can result in reperfusion injury, which may further harm brain tissue. During reperfusion, activated glial cells, injured neurons, and other elements of the neurovascular unit release a mix of substances, including growth factors, matrix metalloproteinases, reactive oxygen species, glutamate, nitric oxide (NO), and cytokines. These molecules contribute to post-injury processes like angiogenesis, breakdown of the blood–brain barrier (BBB), degradation of the basement membrane (BM), glial activation, and neurotoxicity. Moreover, peripheral immune cells infiltrate the brain, collaborating with activated glial cells to foster a pro-inflammatory environment [103,104,105,106,107]. The general features of reperfusion injury are illustrated in Figure 1.

The sub-acute phase of ischemic stroke, spanning from two days to six weeks post-injury, marks the beginning of the repair process. During this phase, cerebrovascular plasticity reaches its peak, and many mediators of vascular injury begin to assume neuroprotective and regenerative roles. VEGF is crucial for post-ischemic neurovascular remodeling. However, the therapeutic effects of VEGF depend on its route and timing of administration; systemic delivery during the acute phase can disrupt the BBB and exacerbate brain injury, whereas local modulation or systemic delivery during the sub-acute phase may mitigate ischemic damage. After an occlusion, BMECs start to proliferate and form sprouts within 12 to 24 h, leading to new capillary formation within three days. Brain pericytes also play critical roles in stabilizing microvessels and promoting neuroprotection following stroke [108,109]. A key characteristic of the sub-acute phase following stroke is the formation of a regenerative neurovascular niche. In this niche, angiogenic blood vessels signal neural progenitor cells, promoting neurogenesis. This is demonstrated by the presence of newly formed neurons near remodeled blood vessels in the rodent striatum and cerebral cortex post-stroke, likely driven by the recruitment of neural stem cells by BMECs, which create a supportive environment. As the stroke recovery enters the chronic phase, starting around three months post-event, the endogenous plasticity diminishes. However, vascular repair, angiogenesis, and behavioral recovery continue, albeit at a slower rate [110,111,112,113]. As said, during the chronic phase of stroke, the formation of new blood vessels, known as angiogenesis, is believed to play a role in brain plasticity and functional recovery. Research, including MR imaging studies in laboratory animals, indicates that the early increase in CBV may result from improved collateral flow, while a later increase is likely due to angiogenesis [114]. Angiogenesis is a critical restorative mechanism triggered by ischemia, occurring not only in the brain but also in the heart and hind limbs. Studies in animal models have shown that promoting angiogenesis correlates with reduced injury [115]. In humans, post-mortem analysis of brain tissue from ischemic stroke patients reveals angiogenic activity, particularly in the penumbral regions surrounding the infarct, suggesting that angiogenesis in these areas could be beneficial for brain recovery [116,117]. A positive correlation has been found between the presence of new vessels in the ischemic penumbra and prolonged survival, indicating the potential advantages of activated angiogenesis for the ischemic brain. However, angiogenesis is not always observed in the peri-infarct area, and factors such as aging can significantly affect post-ischemic angiogenesis. For instance, hypoxia-induced angiogenesis in the rodent hippocampus decreases with age, which is crucial to consider since ischemic stroke predominantly affects older adults [118,119].

The main features of the stages of a stroke are summarized in Table 4.

Scanning electron microscopy (SEM) of vascular corrosion casts from ex vivo brain samples can be used to assess the three-dimensional structure and survival of newly formed blood vessels. Using this technique, it was found that neocortical arterioles and venules lost their radial patterns shortly after occlusion, quickly reforming into a dense network of anastomosing microvessels, similar to normal brain vasculature but distinct from those in growing tumors [120].

Post-ischemic angiogenesis is associated with increased CBF and CBV [114,121]. Noninvasive MRI techniques can monitor vascular remodeling and quantitatively assess tissue perfusion and microvascular characteristics after stroke. These techniques include dynamic susceptibility contrast-enhanced (DSC) MRI, steady-state contrast-enhanced (ssCE) MRI, and arterial spin labeling (ASL). Vessel size index (VSI) and microvessel density (MVD) can also be assessed using ssCE-MRI [122,123]. However, increases in CBF and CBV are not exclusively indicative of neovascularization and may also result from vasodilation or collateral flow. Additionally, BBB disruption can complicate the analysis of MRI data, leading to potential misinterpretation. In one study, despite the proliferation of endothelial cells in the ischemic striatum, ssCE-MRI detected a significant decrease in regional CBV (rCBV) in small vessels, which the authors attributed to cystic transformation, a process that may obscure true increases in vessel density [124]. While MR angiography offers valuable insights, its spatial resolution is limited, making it challenging to directly visualize newly formed microvasculature, particularly in small animal models. Histological staining, often combined with imaging techniques, is therefore commonly used in experimental studies to detect new vessel formation. Recently, high-resolution synchrotron radiation X-ray angiography has been developed, allowing for the imaging of microvessels with diameters in the micron range, enabling direct comparisons of microvessel density before and after ischemia [125].

Following a stroke, angiogenesis and tissue remodeling are particularly active in the penumbral regions surrounding the infarct core. The penumbra is resilient and rapidly initiates recovery processes, including angiogenesis, which plays a critical role in subsequent neurogenesis, vascular remodeling, and overall functional recovery. Endothelial cell proliferation begins as early as 12–24 h after focal cerebral ischemia and can persist for several weeks, with microvascular ECs secreting growth factors and chemokines that support the survival of newly formed neurons [126].

Angiogenesis, neurogenesis, and synaptic plasticity are endogenous processes that occur under both normal and pathological conditions. These processes can also be stimulated pharmacologically. For example, administering human cord blood-derived CD34+ cells post-stroke has been shown to induce revascularization in the peri-infarct cortex and enhance neuroblast migration toward the damaged tissue. Similarly, transplantation of EPCs has been found to promote angiogenesis and improve long-term stroke outcomes in mice [127].

Interestingly, common pathways are activated after stroke during different recovery periods, with some, like the metalloproteinase (MMP) family and chemokine stromal-derived factor-1 (SDF-1 or CXCL12), displaying a biphasic nature. These factors, initially associated with detrimental effects in the early phase post-stroke, later play crucial roles in neuroblast migration and neurorepair during the remodeling phase [128]. Understanding these pathways better could lead to new therapeutic strategies for treating ischemic stroke.

### 3.6. Microvascular Issues in Neurodegenerative Diseases

Cerebrovascular changes are intricately linked with neurodegenerative diseases like Alzheimer’s disease (AD) and Parkinson’s disease (PD). These changes, including BBB dysfunction, hypoperfusion, and altered angiogenesis, might contribute to or exacerbate neurodegeneration. Understanding the sequence and impact of these vascular changes is crucial for developing therapeutic strategies. The complex relationship between cerebrovascular changes and neurodegenerative diseases goes beyond what is observed during normal aging. It is not yet fully understood whether these cerebrovascular changes precede, follow, or occur concurrently with neurodegeneration, but accumulating evidence suggests that these changes can occur before the presentation of symptoms and may even promote neurodegeneration [14,129,130,131,132,133,134]. Table 5 summarizes the main key points of this association, including dedicated issues to AD and PD.

Neurodegenerative diseases involve the gradual loss of neurons and deterioration of brain function. Aging is the most significant risk factor for these diseases, which share some underlying pathological mechanisms with normal brain aging, such as oxidative stress, mitochondrial dysfunction, and the accumulation of toxic proteins [135].

Despite the differences between various neurodegenerative diseases in terms of risk factors, brain regions affected, and symptoms, several common cerebrovascular changes are observed [136]: (I) BBB dysfunction; (II) cerebral hypoperfusion and glucose hypometabolism; (III) dysregulation of VEGF, either overexpressed (as seen in Alzheimer’s, Parkinson’s, and Huntington’s diseases) or underexpressed (as in amyotrophic lateral sclerosis, ALS), affecting blood vessel formation and maintenance.

These vascular changes are crucial because they might not only reflect the disease process but also drive it, potentially offering targets for therapeutic intervention [137,138]. AD is strongly associated with a range of cerebrovascular changes, including vascular degeneration, altered angiogenesis, reduced cerebral blood flow (CBF), increased BBB permeability, and impaired neurovascular coupling. According to the amyloid hypothesis, the buildup of amyloid-beta (particularly the Ab-42 isoform) in the brain results in neurodegeneration due to a toxic gain of function. Furthermore, AD is linked to cerebral amyloid angiopathy (CAA), in which the Ab-40 isoform primarily accumulates around blood vessels in the leptomeningeal and cortical areas of the brain. The connection between cerebrovascular risk factors and the progression of AD is well-established [139,140]. Post-mortem analyses of AD brains reveal numerous vascular abnormalities, including loss of capillaries, atrophied BMECs and pericytes, swollen astrocytic end-feet, hypercontractile SMCs, thickened BM, and BM deposits [78,140]. In severe cases of CAA, additional complications like SMC loss, duplicated vessel lumens, fibrinoid necrosis, and hyaline degeneration are observed. Interestingly, despite capillary loss, AD brains show elevated levels of VEGF, which correlates with increased angiogenesis, neurofibrillary tangles, and amyloid-beta load [141,142]. In transgenic AD models, such as aged Tg2576 mice, tight junction disruptions have been associated with increased microvascular density, suggesting that amyloid-beta may drive both hypervascularization and BBB breakdown in early AD stages, followed by microvascular degeneration in later stages [143]. However, the role of amyloid-beta in brain angiogenesis remains controversial, as it has been reported to both stimulate and inhibit angiogenesis in various studies [144,145].

Reductions in CBF, heightened BBB permeability, and decreased glucose metabolism often precede neurodegeneration in AD, as seen in both animal models and human studies. These changes are evident in individuals at high risk for AD, such as those carrying the APOE ε4 allele or those with mild cognitive impairment (MCI), and persist throughout AD progression, correlating with the severity of cognitive decline [146,147]. Neutrophil adhesion in brain capillaries has also been implicated in reducing cortical blood flow in AD models. Additionally, increased BBB permeability has been observed before plaque formation in AD models, and MRI studies have shown that BBB permeability correlates with cognitive decline in MCI patients. Positron emission tomography (PET) imaging further demonstrates that reduced glucose metabolism, an early marker in high-risk AD patients, precedes brain atrophy [148].

Altered CBF, disrupted metabolism, and compromised blood–brain barrier (BBB) integrity contribute to ischemic and hypoxic damage to the neurovascular unit, which may trigger or exacerbate AD pathology. Impaired neurovascular coupling is commonly observed in AD patients, carriers of the APOE ε4 allele, and transgenic AD mice. Interestingly, treatments that restore neurovascular coupling have been associated with cognitive improvements. Changes in vascular reactivity factors, such as endothelial nitric oxide synthase (eNOS) and endothelin-1, may drive this neurovascular uncoupling and chronic hypoperfusion in AD. Notably, studies have shown that partial eNOS deficiency in mice can lead to CAA-like pathologies and stroke in brain regions suffering from hypoperfusion, closely mimicking conditions seen in preclinical AD patients [149,150,151].

Another interesting issue is related to the role of the amyloid beta (Aβ) peptide in angiogenesis, which is frequently reported in the brain of AD patients in addition to neurodegeneration and neuroinflammation [152]. This is due to the toxicity of Aβ peptide towards neurons and vascular cells (EC and SMCs) [153,154]. The consequent hypoxia and hypoperfusion together with neuroinflammation, triggered by Aβ peptide, provoke a pathological angiogenesis [155]. This last one increases the inflammatory process through the extravasation of blood components and adds this effect to the direct induction of inflammation by Aβ. It has been proposed that Aβ can directly trigger angiogenesis, as demonstrated by the early documentation of angiogenesis in animal models of AD, regulating this process through placental growth factor (PlGF) and angiopoietin 2 (AngII) expression [156]. Moreover, vascular dysfunction in AD can be induced and worsened by arterial hypertension though the Aβ peptide deposition and vascular toxicity [157]. Interestingly, a recent study demonstrated that immunization with Aβ peptides reverses hypervascularity in Tg2576 AD mice and resolves plaque burden. This is an indirect suggestion that neoangiogenesis could be a relevant mechanism underlying plaque formation [158].

PD is associated with a range of cerebrovascular changes, including both vascular degeneration and angiogenesis. Histopathologically, PD is defined by the loss of dopaminergic neurons in the substantia nigra, a region in the brainstem. Additionally, Lewy bodies—aggregates primarily composed of α-synuclein—form within the cytoplasm of certain neurons across various brain regions. The failure to adequately degrade α-synuclein as people age is believed to contribute to the loss of dopaminergic neurons, leading to the motor deficits (like bradykinesia) characteristic of PD [159].

The vascular degeneration observed in PD shows similarities to that seen in AD, with morphological changes such as degeneration of BMECs and a reduction in microvascular density. However, some studies report an increase in angiogenic vessels, indicated by the expression of integrin αvβ3, in brain regions affected by PD, despite no significant change in overall microvascular density. This suggests that a proangiogenic environment might develop in PD, as indicated by elevated levels of angiogenic biomarkers [160,161]. In animal models of PD, integrin αvβ3 expression has been linked with BBB leakage, and post-mortem analysis of PD brain tissue shows evidence of BBB disruption, particularly in the striatum. This implies that angiogenesis may contribute to BBB disruption and, consequently, to vascular degeneration in the later stages of PD. Moreover, reduced blood flow in specific brain regions correlates with the severity of motor dysfunction in PD patients [162,163]. Interestingly, increased microvascular plasticity has been observed in PD patients who undergo deep brain stimulation (DBS). In these patients, reductions in microvascular density, the expression of endothelial tight junction proteins, and endothelial VEGF expression, which are typical of PD, were reversed. DBS appears to enhance neural activity, which in turn rapidly boosts the production of neurotrophic and angiogenic factors like VEGF and brain-derived neurotrophic factor (BDNF) [164]. This suggests that DBS may play a role in restoring cerebrovascular integrity and function in PD patients [165].

### 3.7. Neurovascular Niche and Neurogenesis

Over the past decade, research has revealed the crucial role of blood vessels in regulating adult neurogenesis by providing structural and biochemical support for neural stem cells (NSCs). Blood vessels in neurogenic regions like the subventricular zone (SVZ) facilitate NSC maintenance, proliferation, differentiation, and migration via EC-derived signaling molecules [166]. The vasculature in the SVZ is distinct with a partially permeable BBB and unique interactions between NSCs and ECs mediated by factors like CXCL12 and integrins. ECs also maintain NSC quiescence through signaling molecules like EphrinB2, Jagged1, and gap junctions formed by connexin 43 [167]. ECs release factors such as VEGF, PEDF, and BDNF to promote neurogenesis and NSC proliferation. Blood vessels also provide scaffolding for neuroblast migration. Pericytes, though less understood, play a role in NSC regulation and blood vessel function. Aging leads to significant vascular changes, impairing neurogenesis through factors like increased vascular cell adhesion molecule 1 (VCAM1) and inflammatory signals [168,169].

The CSF, produced by the choroid plexus (ChP), also influences NSC behavior, providing growth factors and other molecules that regulate NSC quiescence, proliferation, and migration. Age-related changes in the ChP and CSF reduce neurogenesis, but rejuvenation strategies like the infusion of young CSF show the potential to reverse some effects of aging. Key molecules involved include IGF1, BMP5, and FGF17. Age-related inflammatory signals, like interferon, negatively impact neurogenesis, and blocking these signals can restore some regenerative capacities in the aging brain [170].

Adult hippocampal neurogenesis (AHN) within the subgranular zone (SGZ) of the dentate gyrus (DG), comparing it to the subventricular zone (SVZ) in terms of structure, cellular components, and regulatory mechanisms is summarized in Table 6.

## 4. The Glymphatic System and Neuroimaging Perspectives in Humans

The biological mechanisms underlying the cerebral microvascular component in lifelong plasticity, in health, and in multiple pathological conditions are known and mostly inferable from animal models and in vivo biological studies with different methods. Studies that can be performed on humans are much less informative from this point of view by their nature, given the impossibility of a dynamic in vivo study of the neurovascular unit and cerebral small vessels. On the other hand, the histopathological study, which represents the pathological gold standard for most brain diseases, is only rarely available and in any case does not allow for following a phenomenon in a dynamic manner, representing a snapshot in a phase, often final, of a pathological process. Recently, with the development of the glymphatic system hypothesis as a unifying pathophysiological theory of microvascular function on both the arteriolar-capillary and venular sides, some further developments have led to the formulation of new hypotheses based on data obtained on humans with non-invasive or minimally invasive study methods, particularly from a neuroradiological point of view, especially with MRI technique, recently also covering, albeit in theoretical terms, phases of development [172]. This new perspective is directly focused on the microcirculation, particularly the cerebral one, providing a potential window on angiogenesis (and neoangiogenesis) and embryonic vascularization with both diagnostic and therapeutic implications. In fact, a significant amount of research has explored angiogenesis, primarily using animal models and various visualization techniques like MRI, micro-CT, and histology [173,174]. Numerous contrast agents are available, each with specific applications, making it challenging to select the most suitable one for imaging (micro)vascular structures [175]. Micro-CT is particularly valuable for visualizing embryonic vasculature, offering superior resolution and efficiency compared to MRI, with images in the micrometer range. However, the effectiveness of micro-CT depends on the careful selection of contrast agents, as each has distinct pros and cons. For optimal soft tissue and vascular imaging with micro-CT, contrast-enhancing staining agents (CESAs) are essential.

The glymphatic system, a network of perivascular spaces (PVSs) surrounding the brain’s blood vessels, plays a crucial role in directing CSF into the brain’s parenchyma, supporting fluid homeostasis, brain waste clearance, and cognitive function. Previously, CSF was mainly believed to protect the brain by providing buoyancy, but recent findings reveal the glymphatic system’s vital role in removing neurotoxic waste and interstitial fluid. PVSs, also known as Virchow–Robin spaces, vary structurally along the brain’s vasculature, with astrocyte endfeet being essential to facilitating glymphatic flow. These astrocytes express Aquaporin-4 (AQP4) proteins, which help mediate the flow of CSF into the brain, though the exact mechanism remains unclear [176,177,178,179,180,181]. The glymphatic system, often referred to as the central nervous system’s lymphatic system, plays a critical role in efficiently exchanging CSF and interstitial fluid (ISF) to clear waste from the brain. This process begins with CSF entering periarterial spaces from the subarachnoid space, driven primarily by arterial pulsations. Astrocytes, a type of glial cell, are central to this exchange, as their vascular endfeet form perivascular spaces around arteries, enabling the smooth flow of CSF into the brain’s interstitial space. A vital component of this system is the water channel protein AQP4, abundantly expressed in astrocyte endfeet. AQP4 channels facilitate the entry of CSF into the brain parenchyma, where it mixes with ISF. The combined fluid then exits the brain through perivenous spaces via diffusion and convection, promoting the clearance of waste [175,176,177]. This cyclical flow helps clear interstitial solutes, including waste proteins like amyloid-β, from the CNS [182,183,184]. These waste products are then drained through several pathways, including meningeal and cervical lymphatic vessels, nasal lymphatics, dural sleeves of cranial and spinal nerves, and transvenous routes via arachnoid granulations. The term “glymphatic system” highlights its dual role, combining “glia” (referring to astrocytes) and “lymphatic” to emphasize its function in clearing brain waste. Understanding of this system has advanced significantly, thanks to real-time two-photon imaging studies conducted on rodent brains [185].

MRI and positron emission tomography/computed tomography (PET/CT) are essential tools in both foundational research and clinical studies for non-invasively exploring the glymphatic system [186]. MRI, in particular, is highly effective for studying the glymphatic system, providing a detailed 4D view of CSF flow dynamics, especially in humans. Advanced MRI techniques have been developed to investigate the glymphatic system in both neurophysiological and pathological contexts [179,187,188,189,190,191,192]. MRI is the primary tool for visualizing CNS fluid dynamics, particularly in relation to the glymphatic system. It enables researchers and clinicians to track changes in glymphatic pathways, such as those influenced by sleep or neurodegenerative diseases, and to examine factors that affect glymphatic flow. A key advancement in this area is the use of contrast-enhanced T1-weighted MRI, which employs gadolinium-based contrast agents (GBCAs) to map glymphatic pathways. These agents, administered either intrathecally or intravenously, enhance the visualization of CSF movement and suggest that gadolinium may enter the brain via the glymphatic system, interacting with both the blood-CSF and blood–brain barriers [193,194,195,196]. Additionally, diffusion MRI techniques, particularly diffusion tensor imaging (DTI), have significantly advanced the understanding of the glymphatic system without requiring GBCAs. DTI tracks the movement of water molecules in three dimensions, offering valuable insights into the glymphatic pathway under various conditions, from chronic illnesses to acute injuries. By quantifying diffusivity, DTI provides a detailed overview of the system’s functionality, helping researchers and clinicians assess changes in fluid dynamics and glymphatic flow more comprehensively [197,198,199,200]. GBCAs [201] are crucial in studying the glymphatic system due to their ability to enhance MRI images by shortening the T1 relaxation time of surrounding water molecules in the brain, thereby improving the visibility of the glymphatic pathways. These agents, when introduced intrathecally or intravenously, allow researchers to visualize CSF movement within the glymphatic system, which is essential for understanding CSF exchange dynamics and the brain’s waste clearance mechanisms. There are some fundamental issues in contrast to MRI imaging of the glymphatic system:
-Intrathecal Administration of GBCAs [202,203]: Intrathecal administration involves injecting GBCAs directly into the CSF, bypassing the BBB. This method is commonly used in both clinical and research settings, particularly in rodents via intracisternal administration into the cisterna magna. Although intrathecal administration is off-label and carries risks such as encephalopathy and severe headaches, careful administration of small doses (typically 0.5 mmol or less) has been deemed safe. This approach has proven effective in evaluating glymphatic function by observing changes in MRI signal intensity over time, allowing for detailed analysis of CSF dynamics and waste clearance. Contrast-Enhanced T1-Weighted Imaging for Glymphatic Flow [204,205]: Using GBCAs, particularly through intrathecal administration, has been instrumental in assessing glymphatic system function. By monitoring MRI signal changes post-GBCA administration, researchers can visualize and assess the distribution of these agents, thereby evaluating glymphatic flow. This method has been adapted from clinically recognized procedures like myelography and cisternography to specifically study glymphatic function, offering insights into the kinetics and spatial distribution of paravascular CSF-ISF exchange [206,207,208,209,210]. Intravenous Administration of GBCAs [211,212,213,214,215,216,217,218,219,220,221,222,223]: Despite the challenges posed by the BBB, intravenous administration of GBCAs has shown potential for assessing glymphatic function. Research indicates that GBCAs can penetrate the brain, possibly through the choroid plexus, with varying degrees of success depending on the GBCA’s chemical structure and molecular size. While the detection of GBCA-induced T1 shortening in brain parenchyma post-intravenous administration is difficult due to minimal GBCA penetration, advanced MRI techniques, such as T2-weighted fluid-attenuated inversion recovery (FLAIR) imaging, have shown promise in detecting low concentrations of GBCAs in the CSF [224,225,226].

Recent studies have explored the safety and efficacy of GBCAs for imaging glymphatic function. For example, Eide et al. demonstrated that intrathecally administered gadobutrol at doses up to 0.5 mmol is safe and effective for imaging purposes, providing valuable data on glymphatic system function [98,227]. Moreover, studies using dynamic time-series contrast-enhanced MRI have offered insights into glymphatic CSF-ISF exchange, particularly in relation to neurodegenerative diseases like AD.

Noninvasive MRI techniques have been increasingly utilized to assess the glymphatic system in humans [228]. These techniques, including phase-contrast imaging, DTI, ASL, and others, offer the advantage of monitoring CSF dynamics and brain fluid flow without the need for exogenous tracers, thus enabling the study of these processes under natural physiological conditions. Phase contrast imaging is a technique that visualizes fluid movement within the body, including CSF, by employing magnetic gradients in MRI to detect velocity changes in moving spins [229,230,231]. This method allows for the measurement of CSF velocity, making it particularly effective in analyzing CSF dynamics in the brain’s ventricles and subarachnoid spaces. For instance, it has been used to observe pulsatile CSF movements in specific brain regions like the foramen of Monro and the Sylvian fissures. However, using phase contrast methods to assess ISF dynamics within the brain parenchyma presents significant technical challenges. DTI is a diffusion MRI technique that assesses the movement of water molecules in brain tissues, offering insights into ISF dynamics [232,233,234,235,236,237,238]. A specific application, DTI-ALPS (Analysis along the Perivascular Space), evaluates the diffusivity of water along perivascular spaces near the lateral ventricles. The ALPS index, derived from this technique, serves as an indicator of glymphatic system activity. DTI-ALPS has been used to study glymphatic dysfunction in various conditions, such as rapid eye movement sleep behavior disorder (RBD) and Parkinson’s disease, demonstrating its potential as a tool for early diagnosis. However, the technique has limitations, including its restriction to measuring diffusivity along the x-, y-, and z-axes, potential difficulties in isolating diffusivity in regions with complex fiber tracts, and the subjectivity involved in manually placing the region of interest (ROI). ASL is a non-invasive MRI technique that measures cerebral perfusion by magnetically labeling blood water, offering a safer alternative to contrast agents [239]. It is particularly useful in studying blood-CSF barrier function and glymphatic dynamics. Despite its clinical relevance, ASL faces challenges such as a low signal-to-noise ratio and limited spatial specificity. Nevertheless, it has been used effectively to study water transfer at the blood–brain interface, as demonstrated in studies on amyloid-β clearance in AD models. Other noninvasive MRI techniques, including ultra-long echo time, low b-value, and multi-direction diffusion-weighted MRI sequences, have also been explored for assessing glymphatic function [240]. These methods, like the ones mentioned above, offer the advantage of noninvasive monitoring of CSF and ISF dynamics, providing valuable insights into the glymphatic system’s role in brain health and disease. These noninvasive MRI techniques, particularly phase contrast imaging, DTI-ALPS, and ASL have significantly advanced our understanding of the glymphatic system. They provide critical insights into CSF and ISF dynamics under natural physiological conditions, contributing to the study of neurodegenerative diseases and other conditions affecting brain fluid flow. Despite certain limitations, these techniques continue to evolve, offering increasingly sophisticated tools for exploring the glymphatic system’s role in maintaining brain health and in diseases [241].

Cerebral small vessel disease (CSVD) is characterized by the dysfunction of the brain’s small blood vessels, leading to damage in both white and gray matter. CSVD is increasingly prevalent, especially in older populations, and can result in strokes, cognitive decline, and balance issues. Neuroimaging markers like enlarged perivascular spaces (EPVSs) and white matter hyperintensities (WMHs) help identify CSVD. While traditionally attributed to hypoperfusion and hypoxia, recent evidence suggests that CSVD-induced brain damage may also involve glymphatic dysfunction. This review explores the anatomy and physiology of the glymphatic system, examines how CSVD risk factors overlap with glymphatic impairment, and discusses how CSVD may exacerbate brain damage by hindering waste clearance through the glymphatic system. However, the role of glymphatic dysfunction in the initial stages of CSVD remains unclear [242,243].

AD is a leading cause of dementia, primarily affecting older adults and characterized by progressive cognitive decline and memory loss. The prevalence of AD increases significantly with age, and its development is influenced by a combination of genetic, environmental, and lifestyle factors [244,245]. A hallmark of AD is the accumulation of extracellular amyloid-beta (Aβ) aggregates, which form amyloid plaques that contribute to neuronal damage and behavioral changes. Interestingly, Aβ deposits have also been found in the meningeal tissues of AD patients, suggesting a broader distribution of these aggregates in the brain and its surrounding structures [246]. The glymphatic system plays a crucial role in the brain’s waste clearance, including the removal of harmful substances like amyloid-beta. Dysfunction in this system is associated with increased accumulation of amyloid-beta plaques, a defining characteristic of AD. Research has increasingly focused on the glymphatic system’s ability to clear neurotoxins, highlighting its importance in maintaining brain health. Age-related declines in CSF and ISF flow, as well as deteriorating lymphatic vessel function, are thought to contribute to the accumulation of Aβ in brain tissue [246,247,248,249]. However, the potential reduction in meningeal lymphatic drainage with age and its impact on AD pathology remains an area in need of further exploration. Studies using rodent models have provided valuable insights into the glymphatic system’s function in brain waste clearance [184,250]. For instance, Iliff et al. [251] utilized dynamic contrast-enhanced MRI to observe the exchange between CSF and ISF, identifying key influx nodes that could be important in evaluating the risk and progression of AD. In a mouse model, MRI studies have suggested that glymphatic dysfunction may contribute to the accumulation of pathological tau proteins, another marker of AD [250]. These findings underscore the potential role of the glymphatic system in AD pathology. In humans, several studies have shown that AD patients exhibit a significant reduction in the DTI-ALPS (Analysis along the Perivascular Space) index, particularly in comparison to cognitively healthy individuals. This reduction suggests impaired glymphatic function. Additionally, a negative correlation has been observed between the volume of the basal ganglia perivascular space (BG-PVS) and the DTI-ALPS index, indicating that EPVSs) may be linked to glymphatic dysfunction in AD. AD patients also tend to have larger BG-PVS volumes, which further points to disturbances in the brain’s waste removal system. These findings suggest that the DTI-ALPS index could serve as a valuable biomarker for assessing glymphatic function and monitoring AD progression [179,216,240,252,253,254].

The pathophysiological hypothesis underlying the glymphatic system concept, although attractive, has been criticized starting from experimental findings on the retinal transport of solutes [255] and the variations in brain clearance of waste during sleep and anesthesia [256,257].

## 5. Conclusions

The vascular side (and in particular the microvascular side) of brain plasticity has been explored to a limited extent in the literature, privileging the neuronal component of plasticity. In reality, the cellular and vascular components are closely interconnected since the developmental phase and the mechanisms of angiogenesis and neoangiogenesis are regulated in a similar manner throughout life and can also be identified in adulthood, albeit to a limited extent, in response to damage or chronic diseases, both vascular and degenerative, as well as to stimuli and behaviors (hypoxia, exercise, etc.). The study of the vascular side of plasticity is particularly complicated in the human brain, being able to rely on non-invasive methods that are not able to return in detail the characteristics and dynamism of the phenomenon, unlike what is possible in the animal model. Furthermore, understanding the molecular and cellular mechanisms underlying the processes of vascular and neuronal plasticity is critical but their acquisition from experimental designs on cellular and animal models makes them not immediately transferable to humans and in some ways sometimes conflicting with each other. The pathophysiological hypothesis of the glymphatic system as a transport pathway for solutes in the brain has rekindled interest in the development and improvement of MRI study strategies dedicated to these issues, which may in the future provide more precise and sequential information on humans.

## Figures and Tables

**Figure 1 brainsci-14-00983-f001:**
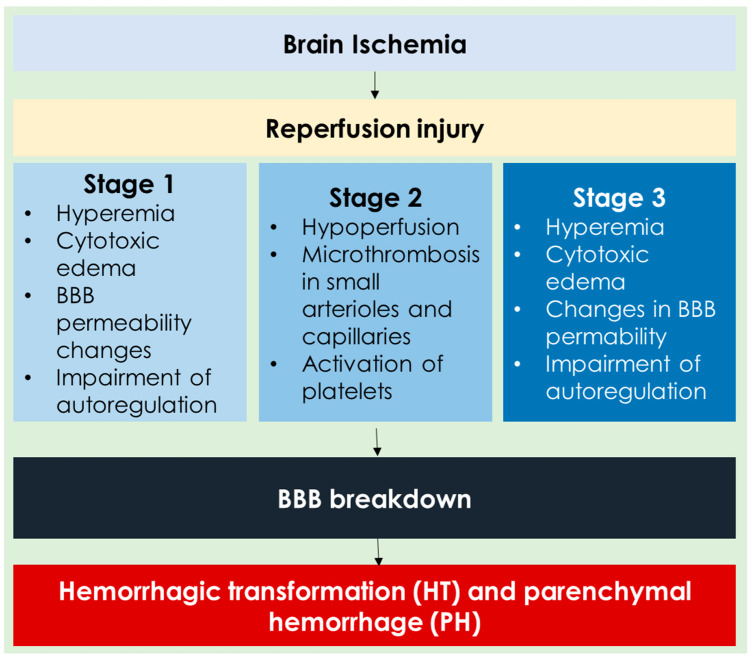
The main steps of reperfusion injury after stroke.

**Table 1 brainsci-14-00983-t001:** Factors influencing cerebrovascular plasticity.

Factor	Main Features
Neural Activity	Local increases in neural activity lead to capillary and arteriole dilation, boosting CBF and the cerebral metabolic rate of oxygen consumption (CMRO_2_). Persistent neural activity changes can prompt cerebrovascular remodeling, as seen in sensory deprivation and hypoxia studies.
Sensory Deprivation	Studies in animals show that stimulating environments can increase capillary density, surface area, and branch points in the brain’s cortex, while sensory deprivation reduces these features. Juvenile and adult animals in enriched environments show faster growth rates and higher microvascular density compared to those in deprived conditions.
Hypoxia	Local and global hypoxia can lead to cerebrovascular changes. Mild hypoxia induces angiogenesis through hypoxia-inducible factor-1α (HIF-1α) and VEGF expression, while severe hypoxia can result in cognitive deficits. Chronic hypoxia in animal models shows regional increases in microvascular density, particularly in the hippocampus and striatum.
Physical Activity	Aerobic exercise promotes neurogenesis and increases microvascular density in brain regions such as the hippocampus and striatum. In stroke models, exercise improves outcomes by increasing perfused microvessels. Exercise-induced cerebrovascular benefits are linked to increased endothelial nitric oxide synthase (eNOS) activity and circulating EPCs. Exercise-induced neuroplasticity is mediated by factors such as VEGF, insulin-like growth factor 1 (IGF-1), and brain-derived neurotrophic factor (BDNF). Exercise raises VEGF levels, promoting neurogenesis in the hippocampus. IGF-1, which increases with exercise, is associated with cognitive improvements and plays a key role in neurogenesis and angiogenesis. BDNF, upregulated by exercise, enhances brain plasticity and cognitive function, with sustained exercise keeping BDNF levels elevated. Blocking IGF-1 or BDNF signaling disrupts exercise-induced neuroplasticity, highlighting their synergistic roles.

**Table 2 brainsci-14-00983-t002:** Impact of hypoxia on the brain [84].

Issue	Main Features
Local hypoxia	In adulthood, local hypoxia is typically associated with injury or disease (e.g., ischemic stroke or the presence of circulating tumor microemboli). These conditions can create localized regions of hypoxia downstream from an occlusion, where the blood supply is reduced or cut off.
Global hypoxia	Changes in environmental oxygen partial pressure, such as at high altitudes, can lead to global hypoxia in the brain. At sea level, where the atmospheric oxygen concentration is 21%, the arterial oxygen partial pressure (PaO_2_) is typically between 75 and 100 mmHg. As altitude increases, atmospheric oxygen concentration decreases, leading to reduced PaO_2_.
Severity	Mild Hypoxia: Characterized by PaO_2_ levels dropping to about 50 mm Hg, which corresponds to an atmospheric oxygen concentration of about 10% or an altitude of approximately 5000 m. Mild hypoxia induces various adaptive responses, including angiogenesis. Moderate Hypoxia: Defined by PaO_2_ levels between 35 and 50 mmHg. It can lead to cognitive deficits. Severe Hypoxia: When PaO_2_ drops below 35 mm Hg, it can result in a loss of consciousness.
Cerebrovascular Responses to Hypoxia	Acute Mild Hypoxia Initial Response: Acute exposure to mild hypoxia causes an increase in CBV due to vasodilation, which is the widening of blood vessels. This also leads to a temporary increase in CBF, potentially up to two times the normal rate. This increase in CBF helps to compensate for the lower oxygen availability. Renormalization: After a few days, CBF typically returns to baseline levels, a process associated with physiological adaptations such as increased RBC volume and higher hemoglobin concentration. These adaptations are well-known benefits of high-altitude training in the field of human performance. Chronic Mild Hypoxia: Microvascular Density: In response to chronic mild hypoxia, such as that experienced over several weeks, rats show region-specific increases in microvascular density. Notably, the hippocampus and striatum exhibit a three-fold increase in microvascular density, while other brain regions display more modest changes. Mechanisms: The increase in microvascular density under chronic hypoxia is driven by angiogenesis. This process is regulated by the expression of hypoxia-inducible factor-1 alpha (HIF-1α), which leads to the upregulation of VEGF, a key driver of new blood vessel formation. Return to Normoxia: Upon return to normal oxygen levels (normoxia), declines in microvascular density occur through the apoptosis (programmed cell death) of BMECs.

**Table 3 brainsci-14-00983-t003:** Impact of aging on NVU and microvascular plasticity.

Issues	Main Features
Microvascular and Neuronal Density	Aging causes a reduction in both microvascular and neuronal density in the brain, particularly in the prefrontal cortex and hippocampus, with microvascular density declining by 10–30%.
Cerebrovascular Plasticity	Compromised with age, alongside neurovascular coupling.
Brain Arterioles	Aging leads to a 40% reduction in arteriolar density, loss of SMCs and elastin, and increases in BM and collagen.
Pericytes and Astrocytes	Aging is linked with degeneration/loss of pericytes, swelling of astrocytic end-feet, and BM thickening.
Tight Junctions	Age-related functional alterations include increased defects in tight junctions and heightened permeability.
BMECs	In aged brains, BMECs show signs of senescence, lower turnover, and reduced response to hypoxia-induced angiogenesis.
Angiogenesis Impairment	Impaired due to factors like hyporesponsiveness of HIF1-α, downregulation of growth factors, inhibition of matrix metalloproteinases, reduced nitric oxide (NO) bioavailability, and pericyte dysfunction.
Interventions	Enriched environments or VEGF delivery through viral vectors improve microvascular density in young and adult rodents, but aging still results in failures in angiogenic signaling, including inadequate upregulation of VEGFR-2.
Regional Alterations	Region-specific changes in angiogenesis-related genes are observed in aged mice.

**Table 4 brainsci-14-00983-t004:** The main stages of reperfusion injury after stroke.

Stage	Processes and Features
Acute Phase	- Lasts up to 48 h after stroke onset. - Oxygen deprivation triggers excessive glutamate release, leading to excitotoxicity and neuronal damage. - Reperfusion injury may occur when blood flow is restored, causing further brain damage. - Activated glial cells and injured neurons release substances like growth factors, matrix metalloproteinases, reactive oxygen species (ROS), glutamate, nitric oxide (NO), and cytokines, contributing to neurotoxicity and BBB breakdown. - Pro-inflammatory environment is formed by peripheral immune cell infiltration and glial activation.
Sub-Acute Phase	- Occurs between 2 days and 6 weeks post-stroke. - Initiation of repair processes, with cerebrovascular plasticity peaking. - Vascular injury mediators become neuroprotective and promote regeneration. - VEGF plays a key role in neurovascular remodeling, though its therapeutic effects depend on timing and administration. - Brain microvascular endothelial cells (BMECs) proliferate and form capillary sprouts. - Pericytes stabilize microvessels and promote neuroprotection. - Formation of a regenerative neurovascular niche that signals neural progenitor cells for neurogenesis.
Chronic Phase	- Begins around 3 months post-stroke. - Endogenous plasticity diminishes, but vascular repair, angiogenesis, and behavioral recovery continue, albeit at a slower pace. - Angiogenesis is a key mechanism in functional recovery, especially in the penumbral regions. - Early increases in cerebral blood volume (CBV) may result from improved collateral flow, while later increases are attributed to angiogenesis. - Factors like aging can negatively impact post-ischemic angiogenesis, especially in hypoxia-induced angiogenesis.

**Table 5 brainsci-14-00983-t005:** Neurodegenerative disease and cerebrovascular alterations.

Issue	Main Features
General issues	Neurodegenerative diseases are marked by progressive neuronal loss and compromised brain function. Age is the greatest risk factor, with common pathologic origins shared between neurodegeneration and normal aging, such as oxidative stress, mitochondrial dysfunction, and proteotoxicity. Despite differences in risk factors, histopathological hallmarks, and clinical manifestations, several themes connect cerebrovascular changes during neurodegeneration: BBB Dysfunction. Cerebral Hypoperfusion and Glucose Hypometabolism. Dysregulated VEGF Expression.
AD	AD is associated with vascular degeneration, reduced CBF, increased BBB permeability, and neurovascular uncoupling. The amyloid hypothesis suggests that amyloid-β (Aβ) accumulation induces neurodegeneration, with cerebral amyloid angiopathy (CAA) contributing to vascular pathology in AD. Pathological changes in AD include capillary loss, atrophied BMECs, thickening of BMs, and BBB disruption. Despite capillary loss, increased VEGF levels are observed, suggesting a complex relationship between Aβ and angiogenesis. Reduced CBF, increased BBB permeability, and reduced glucose metabolism have been found to precede neurodegeneration in AD.
PD	PD is characterized by dopaminergic neuronal loss and the formation of Lewy bodies, with vascular changes resembling those in AD. Evidence of both vascular degeneration and angiogenesis is present in PD, with some studies indicating increased angiogenic vessels and others showing a decrease in microvascular density. Hypoperfusion of specific brain regions in PD patients correlates with motor dysfunction. Treatments like deep brain stimulation (DBS) have been shown to reverse some cerebrovascular changes in PD, potentially through the modulation of neurotrophic and angiogenic factors.

**Table 6 brainsci-14-00983-t006:** Main features of AHN neurogenesis [166,171].

Issues	Main Features
Location and Structure	The SGZ is located in the hippocampus and houses neural stem cells (NSCs) that differentiate into granule neurons. Unlike the SVZ, which interacts with CSF, the SGZ is deep within brain tissue and confined to hippocampal circuits.
Cellular Progression	NSCs in the SGZ progress through stages, starting with radial glia-like cells (RGLs or type 1 cells), which transition into intermediate progenitor cells (IPCs, type 2 cells). IPCs subdivide into type 2a (glial-like) and type 2b (neural-like), eventually forming neuroblasts (type 3 cells) that migrate to the granular cell layer (GCL) for final differentiation.
Blood Vessel Interactions	Blood vessels in the SGZ are more abundant than in the surrounding granule cell layer (GCL), and NSCs interact with endothelial cells (ECs) via signaling molecules like VEGF (vascular endothelial growth factor). Proliferating NSCs are often found near proliferating ECs, suggesting synchronized angiogenesis and neurogenesis. VEGF signaling has been shown to support both angiogenesis and neurogenesis, especially in response to environmental stimuli such as exercise and learning.
Molecular Pathways	VEGF-C/VEGFR3 signaling regulates NSC proliferation and survival, and experiments suggest ECs secrete VEGF-C, potentially acting as an angiocrine signal for NSCs. Blood-borne factors also influence neurogenesis. For example, IGF-1 (insulin-like growth factor 1) increases neurogenesis following exercise, while other factors like GDF11 rejuvenate brain vasculature and boost neurogenesis. Conversely, aging-related factors like CCL11 impair neurogenesis.
Neurovascular Coupling	Recent findings highlight the role of neurovascular coupling, where the activity of pre-existing neurons modulates blood flow and supports neurogenesis. This is mediated by NO signaling, which increases IGF signaling to NSCs.

## Data Availability

Not applicable.

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
