# Peer review of "The Cerebrovascular Side of Plasticity: Microvascular Architecture across Health and Neurodegenerative and Vascular Diseases"

_brainsci, 2024, doi:10.3390/brainsci14100983_

Round 1

Reviewer 1 Report

Comments and Suggestions for Authors

1. Read the whole manuscript for grammatical errors. 

2. Abbreviations like "BMEC" (Brain Microvascular Endothelial Cells), sometimes whole words, and sometimes abbreviation was used, please refer to lines no 111 and 360. 

3. Some of the references were cited in italics, please correct them throughout the manuscript. 

4. Tables are highly dense.

5. No image, figure, or graphical abstract. Please add suitable figures.

6. Improve conclusion part. 

7. Also refer to the blood-brain barrier the terminology is inconsistently used. 

Comments on the Quality of English Language

minor corrections were required 

Author Response

First of all, we would like to thank the reviewer for his/her appreciation of our paper.

1. Read the whole manuscript for grammatical errors. DONE

2. Abbreviations like "BMEC" (Brain Microvascular Endothelial Cells), sometimes whole words, and sometimes abbreviation was used, please refer to lines no 111 and 360. WE CHECKED AND CORRECTED THESE POINTS. 

3. Some of the references were cited in italics, please correct them throughout the manuscript. DONE

4. Tables are highly dense. WE TRIED TO REFORMAT THEM WITHOUT LOSING INFORMATION AND TO USE ACRONYMS

5. No image, figure, or graphical abstract. Please add suitable figures. IT IS NOT A TOPIC SO SUITABLE FOR FIGURES FROM A CLINICAL POINT OF VIEW AND EXPERIMENTAL FINDINGS/FIGURES REQUIRE PERMISSIONS TO BE REPRODUCED. WE COULD ADD SOME NEURORADIOLOGICAL FIGURES, IF THE EDITOR REQUIRES, BUT IN THIS CONTEXT THEY MIGHT BE SOMEHOW OUT OF TOPIC.

6. Improve conclusion part. DONE

7. Also refer to the blood-brain barrier the terminology is inconsistently used. CORRECTED

Reviewer 2 Report

Comments and Suggestions for Authors

This review paper "The cerebrovascular side of plasticity: microvascular architecture across health and neurodegenerative and vascular diseases" is quite rich in textual amount and the topic is certainly of both basic and translational importance. In general, the review can be published, but it would be hard for the readership to grasp the central idea of the review.

I find that the there is a tendency to reiterate well-known facts about cerebrovascular architecture and brain plasticity without contributing new insights or synthesizing existing knowledge innovatively. In the age of artificial intelligence-assisted text composition, review articles should deliver novel perspectives based on existing knowledge.

I find this review is not so easy to read through because it lacks a coherent structure, with sections flowing into each other without a strong narrative thread. For instance, discussions on development, aging, and disease-related changes in cerebrovascular architecture/plasticity are not distinctly separated or compared. The last section, namely, the glymphatic system does not connect well to the previous sections or to the central theme (i.e., the title). A more organized structure would improve readability and comprehension. Perhaps the last section could be dropped and instead, this review article would be more compelling if the authors’ own work and view to the central topic are articulated.

While the coverage of cited literature is broad, the context that this review article delivers is not substantial and lacks mechanistic/molecular insights. For instance, while processes like angiogenesis and vascular remodeling are discussed, the authors fail to describe the underlying molecular mechanisms in detail. Schematic diagrams that illustrate the interactions of mentioned molecules would be very helpful to the readership. For example, the role of VEGF in neurovascular remodeling post-stroke is briefly noted, but neither the downstream signaling pathway nor cellular interactions is explained. Including a more detailed discussion of the molecular mechanisms would provide a more in-depth understanding of cerebrovascular plasticity.

Moreover, the authors do not critically analyze conflicting evidence or controversies in the field. For example, the relationship between amyloid-beta and angiogenesis is mentioned, but the authors do not critically discuss the conflicting data on whether amyloid-beta promotes/inhibits vascular plasticity. Likewise, the final section is dedicated to the glymphatic system, but literature that challenges the validity of the glymphatic system (e.g., by Verkman) is completely ignored.

Lastly, I would repeat that while this review article is publishable, the authors ought to highlight gaps in the current understanding or even propose new hypotheses or future research directions.

Minor (ALL OPTIONAL):

Line 11: energy demanding (no hyphen)

Line 39: must state whether this is in rodents or in humans (or both).

Line 44: endothelial growth quiescence?? (still sounds a bit strange)

Line 96: may want to introduce Wang et al. (2022) 10.1016/j.crmeth.2022.100302 for perpetual fluorescent blood labeling.

Line 113: may want to refer to (a different) Wang et al. (2024) 10.1016/j.celrep.2024.114723

Lines 146-154: Parallel comparisons should be made between the human and the mouse (or rodents). It is a good idea to tabularize the numbers by species.

Lines 163-164: This is a controversial topic. Not entirely sure if the cited reference is centrally addressing the topic. Perhaps  10.1101/cshperspect.a041354  ,  10.1016/bs.pbr.2016.02.001 , 10.3389/fnetp.2023.1162757

Lines 167-168: Walchli et al.? (Ref 40)

Lines 186-187: needs a reference(s).

Glymphatic section (section 4). Make sure that the references are aligned. There may be a misalignment between the reference list and cited numbering. The glymphatic system hypothesis has been challenged since the initial publication of the model. For a balanced view, the authors are recommended to introduce work by Alan Verkman and the recent Miao et al. Nature Neurosci (2024).

Minor (ALL OPTIONAL):

Line 11: energy demanding (no hyphen)

Line 39: must state whether this is in rodents or in humans (or both).

Line 44: endothelial growth quiescence?? (still sounds a bit strange)

Line 96: may want to introduce Wang et al. (2022) 10.1016/j.crmeth.2022.100302 for perpetual fluorescent blood labeling.

Line 113: may want to refer to (a different) Wang et al. (2024) 10.1016/j.celrep.2024.114723

Lines 146-154: Parallel comparisons should be made between the human and the mouse (or rodents). It is a good idea to tabularize the numbers by species.

Lines 163-164: This is a controversial topic. Not entirely sure if the cited reference is centrally addressing the topic. Perhaps  10.1101/cshperspect.a041354  ,  10.1016/bs.pbr.2016.02.001 , 10.3389/fnetp.2023.1162757

Lines 167-168: Walchli et al.? (Ref 40)

Lines 186-187: needs a reference(s).

Glymphatic section (section 4). Make sure that the references are aligned. There may be a misalignment between the reference list and cited numbering. The glymphatic system hypothesis has been challenged since the initial publication of the model. For a balanced view, the authors are recommended to introduce work by Alan Verkman and the recent Miao et al. Nature Neurosci (2024).

Author Response

First of all, we would like to thannk the reviewer for his/her observations.

The main purpose of this review is to provide the point of view of the clinician (neurologist, neuroradiologist) on a complex topic, usually addressed from a biological point of view. This last approach is usually very detailed in molecular mechanisms and description of aminal models and experimental designs but lacks of immediate transferability in the clinical reasoning on people and patients. Our aim was to propose a different approach, not focusing on molecular mechanisms and on the comparison between rodents and hymans, but proposing connections between hypothesis and clinical practice. This is the main reason why we choose to not add figures about the detailed cellular and molecular pathways and not propose the challenge about several hypotheses. The structure of te paper reflects this view and the sections are balanced according with this view.

We respectfully disagree about the lack of immediate link between title and sections, in particular the last one. The glymphativ hypothesis is involved in mechanisms of aging, vascular and neurodegenerative diseases. The positions challenging it, added at the end of the session, are mainly derived from single experiments and it is not so clear if and how these fundings migh apply on humans.

We added the references suggested by the reviewer and changed some passages in order to better clarify the text.

Endothelial growth quiescence refers to non proliferating endothelial cells (https://www.nature.com/articles/s41569-021-00517-4). We added this reference too.